# Preconception Care for Men and Women during the Pandemic, an Intervention Proposal

**DOI:** 10.3390/healthcare9070816

**Published:** 2021-06-28

**Authors:** Nieves Estrella Rovira-Vizcaíno, Jesús Sáez-Padilla, José Manuel Romero-Márquez, María de los Ángeles Merino-Godoy

**Affiliations:** 1Regional Hospital La Axarquía, 29700 Málaga, Spain; estrella_22j@hotmail.com; 2Integrated Didactics Department, University of Huelva, 21071 Huelva, Spain; 3Department of Nursing, University of Huelva, 21071 Huelva, Spain; josemr1998268@hotmail.com (J.M.R.-M.); angeles.merino@denf.uhu.es (M.d.l.Á.M.-G.)

**Keywords:** pandemic, preconception, community intervention, lifestyle

## Abstract

The COVID-19 pandemic and its measures resulted in limited outdoor activities, reduced group meetings, etc., leading to unhealthy habits. Several studies showed how certain unhealthy habits can lead to serious consequences for both men and women, as well as affect future offspring. (1) Background: Therefore, we present a community intervention at the preconception stage to avoid future risks. The purpose of this intervention is to change lifestyles and beliefs about the health of men and women in the preconception period; (2) Methods: For the design of the intervention, a bibliographic search was performed both in English and Spanish in the main databases of health sciences and nursing (Cochrane, PubMed, Web of Science, CINAHL, LILACS, Dialnet), using descriptors in MeSH health for sciences; (3) Results: We proposed that a variety of lifestyles be analyzed, including aspects such as physical activity, nutrition, etc. In addition, stress management should be emphasized through a relaxation workshop, where three different techniques be proposed to reduce anxiety levels in stressful situations; (4) Conclusions: Due to the limited scientific results of interventions carried out in the preconception period simultaneously with men and women, more community interventions that address this topic are needed to assess the impact of these actions on the health of the population.

## 1. Introduction

The measures taken during the COVID-19 pandemic limited normal daily events including social distancing, limited outdoor activities, reduced group meetings, etc. This developed so that it became more difficult to carry on with health care education [1].

The period of preconception is the stage prior to pregnancy, so when carrying out interventions at this moment in the lives of women and their respective partners, it is possible to modify the health of both women and their family during pregnancy and the phases to come after this physiological state [2]. Preconception care aims to ensure that couples follow healthy lifestyle habits before conceiving a child, obtaining satisfactory reproductive results [3].

During the preconception visit, according to the Spanish Society of Gynecology and Obstetrics [4], health professionals must achieve the purpose of determining the mother’s health condition prior to pregnancy, recognizing possible hereditary diseases in the parents. Likewise, their role will be to explain the impact of the exposure to certain risk factors on fetal health (toxic habits, substance abuse, etc.) and carry out interventions to prevent neural tube damage in the fetus.

It is necessary to implement health education and promotion activities, intervene in the individual needs of couples and assess their circumstances at home (family relationships, financial situation, etc.). Upadhya et al. [5] explain that preconception care includes the purpose of identifying and modifying those risk factors that can harm maternal–fetal health. Figure 1 presents the three main determinants that will influence the preconception health of the mother and her partner [6].

As a result, preconception care reduces perinatal mortality and morbidity [7], evaluating at the same time the health condition of both the woman and her partner and distinguishing the risk groups that are present in the community, in order to intervene in them [8]. Therefore, preconception care should not only be provided to women who are actively seeking pregnancy but also for those who want to avoid unwanted pregnancies [9] and their partners, as their health status will influence their own physical and mental health, the quality of the sperm, their adaptation to parenthood, and the health of the partner and the offspring [7].

The factors that we must take into consideration when carrying out an intervention with couples are the following: eat a balanced diet, maintain physical activity, exclude toxic habits (tobacco, alcohol, and substance abuse), avoid stress, sleep, and control scrotum temperature.

A discrepancy between diet and physical activity causes obesity in the long term and, nowadays, this, together with an unfavorable lifestyle, increases infertility in our society. People consume more high-fat meals and reduce their level of physical activity, which can influence the development of a greater number of complications in the health of women of childbearing age [10].

Obesity does not only reduce the possibility of future pregnancy for women, but is also related to different risks, such as complications in pregnancy, premature births, or future obesity in their offspring [11]. Regarding the impact of obesity in men, those who suffer from this disease have a lower level of testosterone, higher level of estrogen, higher risk of erectile dysfunction, poorer sperm quality and, therefore, reduced fertility [12].

The key recommendations to consider in preconception interventions, in relation to the nutrition habits of the couple, are the following:Limit the energy consumption that comes from added fats and sugars [10].Increase the intake of fruits, vegetables, legumes, and whole grains [10].Avoid sugary and carbonated beverages, drinking water instead [10].Avoid cooking fried foods and trans fats [12].Cook with vegetable-based oils, such as extra virgin olive oil. It is also important to add nuts (raw, unprocessed) and fish to the diet [12].

Physical inactivity together with obesity is associated with an increased risk of negative obstetric outcomes, such as venous thromboembolism in pregnancy, excessive weight gain, gestational diabetes, and even fetal macrosomia [13]. Exercise helps to improve the physical condition, reduces stress, and improve mental health [14].

When analyzing the positive effects of exercise during pregnancy, benefits are seen for both the mother: a reduction in the percentage of caesarean sections, faster return to post-pregnancy physical condition during postpartum, and better maternal cardiovascular health; and the offspring, as an improved response to stressful stimuli and the reduction of neonatal fat mass [13]. Finally, the scientific evidence reveals that maintaining moderate physical activity and following a balanced diet can help improve men’s fertility [15].

In reviewed studies, we have found negative factors that can affect health, such as when fathers smoke during preconception compared to when mothers carry out this habit. The behavior of smoking fathers was linked with an increased risk in the development of childhood leukemia in offspring, while, in the case of smoking mothers, no relationship was found with the development of childhood leukemia in the offspring [16].

The consumption of steroids, marijuana, alcohol, methamphetamine, and opioids can affect the fertility of both men and women [17]. Likewise, substance abuse in the mother is related to complications during pregnancy, such as increased risk of infant mortality, low birth weight, and neonatal withdrawal syndrome [16].

In mothers, it is noticeable that stress increases the risk of anovulation and creates a hostile environment at the time of implantation [18]. Furthermore, the exposure of women to this pressure during the preconception period is associated with sleep disturbances in the offspring [17]. Creating a sleep routine before pregnancy is a significant change in behavior to reduce stress and thus achieve a better quality of rest [17].

In contrast, stress affects sperm quality in males, developing abnormalities in morphology [12]. It also influences male fertility by decreasing the level of testosterone, modifying the process of spermatogenesis, and even leading to erectile dysfunction or ejaculation problems [18].

Elevation of the scrotal temperature decreases the quality of the sperm. This can happen in several daily situations, such as in a bath or in a jacuzzi, after an episode of high fever, after wearing cycling tights or very tight underwear, when putting the laptop on their lap, and even in working conditions where men are exposed to high temperatures or remain seated for a long period of time [12].

The main objective is to establish healthy lifestyle habits in couples during the preconception stage through educational sessions in primary care, carried out by the multidisciplinary team. Among the specific objectives to work we have: to encourage the need for the care of health habits in men and women in the preconception stage and motivate them both equally in preconception care, strengthening their relationship of responsibility both in pregnancy and the later stages of fatherhood and motherhood.

## 2. Materials and Methods

For the design of the intervention, a bibliographic search was performed both in English and Spanish in the main databases of health sciences and nursing (Cochrane, PubMed, Web of Science, CINAHL, LILACS, Dialnet), using descriptors in MeSH health sciences. The research was limited to publications with a date of no more than 10 years, except for publications useful for the theoretical framework of the sessions. The keywords used in the search were: “preconception”, “preconception care”, “preconception health”, “men’s health”, “couples”, “primary care”, “family planning” and “pre-pregnancy”.

In addition, didactic resources provided by the Spanish Society of Gynecology and Obstetrics, the Public Health England website, the Spanish National Institute of Statistics, the Spanish Ministry of Health, Social Services and Equality, and the World Health Organization were reutilized.

A specialist team in health will design the community intervention. The approach to be followed will be participatory and that, through different activities (workshops, video viewing, debates, etc.), the relevant content of each session would be transmitted.

### Target Population of the Intervention

This intervention is proposed for all couples of women and men who want to improve their reproductive health by receiving care before pregnancy or guidelines to improve their health habits. The optimal number of participants is estimated to be between 4 and 7 pairs.

The main inclusion criteria must be:

Participants will be of legal age. They will present a gestational or adoption desire if they prefer it or are not fertile. Social class is unimportant. Those women and their respective partners who take vitamin supplements and those who do not will be included. People who follow a balanced diet and those who do not take care of their diet. The inclusion criteria will include those couples who have had a previous pregnancy and those who have not (if they have children right now it may also be beneficial to give that health education), and couples who have had a previous abortion.

Exclusion criteria: minors and women who have gone through or are currently experiencing the process of menopause. Pregnant women (not included as we intend to study preconception).

Participants should be recruited through members of the multidisciplinary team of primary care. Likewise, is necessary to emphasize the importance of the nurse case manager being identified as a key link with the hospital so that patients who they consider appropriate can also join this intervention.

Research manuscripts reporting large datasets that are deposited in a publicly available database should specify where the data have been deposited and provide the relevant accession numbers. If the accession numbers have not yet been obtained at the time of submission, please state that they will be provided during review. They must be provided prior to publication. All information will be incorporated into the health database through observation, discussion groups during all sessions, and the final survey in session 7. A selected participant will fill in the written informed consent related to the study in session number one, both the team of professionals and the participants will have a copy of the consent, respectively. This document will reflect the date and signature of the participants, the option to withdraw from participation in the study if the couples so wish. It will be clearly reflected that the option to participate is completely voluntary for each individual capable of deciding for himself. In summary, it will include: the nature of the procedure, the risks and benefits and the procedures/activities to follow, explanation of the benefits and evaluation of the understanding of the patient of everything previously mentioned in the letter.

Intervention studies involving animals or humans, and other studies that require ethical approval, must list the authority that provided approval and the corresponding ethical approval code.

## 3. Results

The intervention consists of seven sessions based on theoretical–practical activities, social interaction, and continuous feedback from the multidisciplinary health team that has the following members:Primary Care Nurse and/or Nurse Specialist.Nurse Specialist in Obstetrics–Gynecology.Medical Specialist in Family and Community Medicine.

It is essential that the health professionals have experience in delivering workshops for health education in preconception, family planning, or women’s health.

First session: What is preconception care? Shared responsibility. Theoretical content on preconception and the impact that different lifestyles have on it must be addressed. The importance of both partners in taking care of their health before pregnancy must be described, as well as the significant role of folic acid and the correct vaccination of future parents.

In this first contact, attendees’ health habits must be analyzed to enable adaptation to these throughout the different sessions. The multidisciplinary team must prepare a healthy habits questionnaire adapting to the specific characteristics of each group of participants.

Second session: Myths and curiosities about fertility and preconception. Myths about sexuality of both men and women must be demystified. Less well-known aspects about different lifestyles carried out that affect fertility must be explained and images of daily activities where these aspects are present must be shown.

Third session: Toxic habits. Tobacco, alcohol, and drug abuse. The impact of harmful health habits on fertility must be exposed. Two groups must be formed, where the first must agree on the consumption of these and the second must have to put into practice the knowledge acquired, creating tips and strategies to try to convince the first group.

Fourth session: Guidelines for a balanced diet and recipes for everyday life.

Strategies on balanced eating and dietary changes that can improve health must be explained. The risk of following an unhealthy diet must also be highlighted. Finally, a recipe book must be distributed with a variety of easy and healthy recipes for everyday use, considering the availability of local and seasonal products, with blank sheets at the end so that participants can create their own meals.

Fifth session: Stress management. Relaxation. The impact of stress on fertility and the different useful techniques for managing stress in pairs must also be useful for practicing individually. Next, the explained techniques must be performed, solving and doubts by the end of this workshop.

Sixth session: Importance of physical activity. *Healthy promenade.* The objective of this session is to make participants aware of the need for doing moderate exercise during preconception. The health team should find a meeting point where participants can take a walk, to encourage regular physical activity. During the tour, a brochure must be given with activities and exercise ideas that can be done in the preconception period.

Seventh session: Program evaluation. In this session the intervention must be globally evaluated. Possible questions of the assistants must be solved, and through a final questionnaire the responsible team must know how the couples feel about the sessions, aspects of improvement that the users would change and what has been useful to them from all the activities.

The sessions will take place over 7 weeks. The following table offers a chronological order to achieve the objectives set progressively, increasing the participation and adherence of the participants (Table 1).

Each session will begin (except for the last evaluation session) by using 5 min so that each couple can write down their motivations or reasons for why they want to change to a health habit and keep these in mind when they doubt whether they can perform changes in their lifestyles, as motivation will make the change more meaningful and add a great responsibility. The assessment period will help us to guide the decisions we make regarding the intervention; it will also indicate if the objectives that we had set for ourselves at the beginning of the intervention have been met.

The assessment must be continuous and final. The continuous evaluation of the intervention will allow us to improve the different aspects of the sessions, while the final evaluation will be the tool that we use to measure the achievement of the objectives and if the couples have acquired the wanted knowledge and skills.

The members of the multidisciplinary health team must write down in a field journal as they carry out the activities, all those suggestions for improvement and testimonies that come from the participants in the last minutes of the sessions dedicated to the sharing of questions and opinions.

As a final evaluation, session number seven will be used for the resolution of doubts and the distribution of a satisfaction questionnaire o analyze the program globally. To study the impact of this community intervention on the participants, two months after the end of these sessions, the nurse of the multidisciplinary team must contact each user, either by phone or email, to check if they are carrying out the new knowledge acquired in their daily life.

## 4. Discussion

One of the consequences of the pandemic is the lack of maternal education. In fact, doctors in all countries of the world are concerned that women do not have the necessary training before, during or after childbirth. In this work, we want to emphasize support at the preconception stage.

Preconception care should be provided to women who are actively seeking pregnancy and those who want to avoid unwanted pregnancies, and their partners, as the health status of both will influence their own physical and mental health, the bio psychosocial aspects of the relationship and the health of the offspring.

The lack of interventions that include both men and women in preconception is evident. We have observed the lack of experiences in Spain, as most of the interventions from other countries make it difficult to apply certain measures to our lifestyles, such as our own customs, beliefs, etc. For example, most of the interventions found on the impact of stress are carried out in animal experiments, finding very few that study this effect on humans, and even fewer during the preconception stage. Moreover, the interventions found that deal with the broader factors that influence preconception, such as mental health and environmental exposure, are limited; this is the same situation for interventions that include LGBTQ groups (lesbian, gay, bisexual, transgender, and queer or questioning).

We propose an intervention that consists of different sessions with a theoretical–practical nature; all of these sessions have been described. We hope to contribute to the health care of future parents and health education in preconception. Our next goal is to develop the designed intervention and assessment to empirically demonstrate its value so that it is endorsed in its bibliographic foundation.

## Figures and Tables

**Figure 1 healthcare-09-00816-f001:**
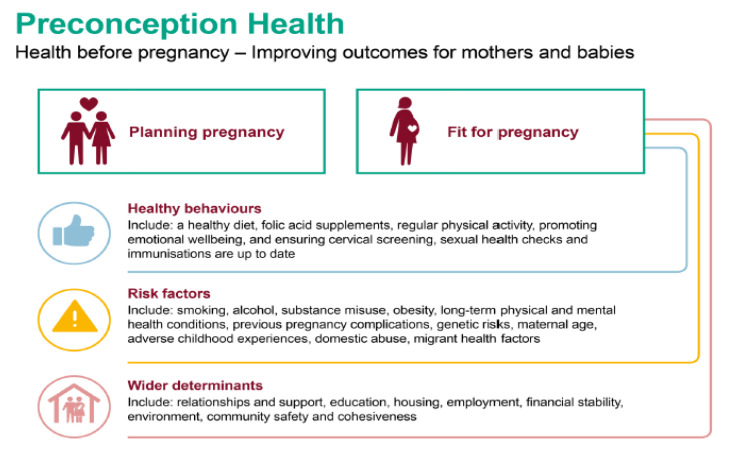
Preconception Health. Influencing factors in the health of mother and children. Public Health England. Planning for pregnancy to improve maternal and child outcomes (2018: 13).

**Table 1 healthcare-09-00816-t001:** Chronogram of the sessions.

Session	Activities	Duration	Week
1st	What is preconception care? Shared responsibility?	40 min	1
2nd	Myths and curiosities about fertility and preconception	40 min	2
3rd	Workshop on toxic habits. Tobacco, alcohol, and drug abuse	1 h	3
4th	Guidelines for a balanced diet and recipes for everyday life	45 min	4
5th	Stress management. Relaxation session	1 h	5
6th	Importance of physical activity. Healthy promenade	1 h	6
7th	Program evaluation	30 min	7

## Data Availability

Not applicable.

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
