# Peer review of "Preconception Care for Men and Women during the Pandemic, an Intervention Proposal"

_healthcare, 2021, doi:10.3390/healthcare9070816_

Round 1

Reviewer 1 Report

I believe that the proposed intervention is appropriate and promising results can be obtained.
However, it would be useful to explain whether the number of participants is adequate, since 4 to 7 couples seems too few to obtain conclusive results.

I don't understand the paragraph from line 97 to line 101, it is contradictory.

A couple of minor corrections:

Abstract: Change on Lines 16, 19 and 22:
period; (2) by period. (2)
sciences; (3) for science. (3)
situations; (4) for situations. (4)

Line 54. I do not understand this reference: (2018: 3)

Author Response

Thank you very much for the corrections

I attach the information of the 2 reviewers

Jesús Sáez

Reviewer 2 Report

The manuscript describes an interesting proposed intervention. This work also provides a brief overview of the current knowledge of preconception care for men and women.

The manuscript reports an interesting prevention intervention proposal through educational sessions in preconception for men and women during the (covid-19) pandemic condition.

Overall, this project is well written and well articulated, with relevant and adequate bibliographic references.

A further in-depth analysis of the text can be carried out.

In response to a request for an opinion on the project, the local Ethics Committee would probably indicate that some elements are missing on how the authors intend to concretize what is proposed within the manuscript.

Therefore, the integration of these elements would certainly also enrich the version of the work published in the Healthcare journal.

Following are the additional questions that should be satisfied.

The inclusion and exclusion criteria of the subjects to be enrolled must be reported in detail.

How will the collection of informed consent take place?

What content do the authors intend to include in the informed consent document?

What are the criteria that will be adopted for the management of the collected data?

How will the privacy of the research sample be guaranteed?

In addition, more information on the tools adopted to evaluate the effectiveness of the interventions is required within the manuscript. Finally, which statistical analyses of the data will be adopted?

I hope that this level of analysis will be useful in increasing the quality of the manuscript.

Author Response

(The authors gave the same response as above.)
